# Proximal and Distal Vagal Indigestion in Buffaloes (*Bubalus bubalis*) in the Amazon Biome

**DOI:** 10.3390/vetsci10040254

**Published:** 2023-03-28

**Authors:** José Diomedes Barbosa, Marilene Farias Brito, André de Medeiros Costa Lins, Camila Cordeiro Barbosa, Paulo Sérgio Chagas da Costa, Marcos Dutra Duarte, Tatiane Teles Albernaz Ferreira, Natália da Silva e Silva Silveira, Carlos Magno Chaves Oliveira, Felipe Masiero Salvarani

**Affiliations:** 1Instituto de Medicina Veterinária, Universidade Federal do Pará, Castanhal 68740-970, PA, Brazil; 2Departamento de Epidemiologia e Saúde Pública (DESP), Instituto de Veterinária (IV), Universidade Federal Rural do Rio de Janeiro (UFRRJ), Seropédica 23890-000, RJ, Brazil

**Keywords:** forestomachs, ultrasound, hoflund’s syndrome, Amazon biome

## Abstract

**Simple Summary:**

The buffalo industry has great importance in the state of Pará in the Amazon biome, as it contributes to society in economic and cultural ways. Thus, it is necessary to know the disorders that affect this species. However, there are no reports in Brazilian literature and only a single article in the international literature on vagal indigestion. Thus, in this work, we describe two different cases of the disease in buffaloes ad present the clinical results as well as complementary exams and macroscopic necropsy findings. A significant finding is chronic distal stenosis in Buffalo 2, which generally occurs in an acute form.

**Abstract:**

This study aims to describe the clinical signs and ultrasonographic and necropsy findings of the first cases of proximal (Buffalo 1) and distal (Buffalo 2) vagal indigestion in two *Bubalus bubalis* in the Brazilian Amazon biome. The clinical histories of the buffaloes were characterized by progressive weight loss, recurrent tympany, abdominal distention (apple and pear shapes), anorexia, and scant feces. Buffalo 1 was submitted to orogastric intubation, and due to the recurrent tympany, an exploratory laparotomy. Buffalo 2 was submitted to ultrasound examination, and a segment of the pylorus was shown to be adhered to the eventration by ultrasonography. Both animals produced positive results for the atropine test. In the necropsy evaluation, Buffalo 1 was shown to have dilation of the esophagus, rumen, and reticulum; the ruminal contents of animal 1 were olive green and foamy with bubbles within the ingesta. On the other hand, Buffalo 2 was shown to have distention of the forestomach and abomasum; the complex rumen–reticulum and omasum contents were semi-liquid and had a yellowish color. In animal 2, in the eventration region, there was adherence to the pyloric region. The diagnosis of vagal indigestion was based on the history, clinical signs, and ultrasound and necropsy findings, in addition to the results of the atropine test.

## 1. Introduction

Vagal indigestion, also known as “Hoflund’s Syndrome” or functional stenosis, is characterized by dysfunction of the tenth pair of cranial nerves, which leads to changes in the motor functions of the forestomach [1,2]. The vagus nerve is not only found in the head region but also projects into the thoracic and abdominal cavities, where it branches, forming a visceral plexus. In the abdominal cavity, the vagus nerve is divided into a dorsal trunk, which innervates the rumen, and a ventral trunk, which is divided into several branches and, when passing through the diaphragm and leaving the esophageal hiatus, innervates the reticulum, omasum, abomasum, and liver—regions where inflammation often appears [1,3]. According to Dirksen [1] and Fubini and Divers [4], lesions of the ventral vagal trunk occur more frequently because they can cause a proximal stenosis, which inhibits the passage of food from the reticulum through the omasum, or a distal stenosis, which inhibits abomasal flow.

Vagal indigestion mainly affects cattle raised in semi-intensive or intensive systems and is rare in small ruminants [5,6,7,8,9,10,11]. In relation to Brazil, one of the largest meat exporters with one of the largest cattle herds in the world, few reports of vagal indigestion are described in the literature [11,12,13]. There are no reports of vagal indigestion in buffaloes in Brazilian literature and only a single article in the international literature [14]. Due to the importance of the disease and the scarcity of reports on the occurrence of this disease in *Bubalus bubalis* in the literature*,* the objective of this study was to describe the clinical signs and ultrasound and necropsy findings of the first cases of proximal and distal vagal indigestion in two buffaloes from the Brazilian Amazon biome.

## 2. Materials and Methods

Two buffaloes (*Bubalus bubalis*) were examined by the veterinary medical team of the Veterinary Hospital of the Institute of Veterinary Medicine of the Federal University of Pará, Campus Castanhal. The animal identified as Buffalo 1 belonged to a property located in the municipality of Nova Timboteua, and the animal identified as Buffalo 2 belonged to a property located in the municipality of Castanhal. Both of these cities are located in the state of Pará, Brazil.

Buffalo 1 was a one-year-old Murrah female (*Bubalus bubalis bubalis*, the river buffalo), weighing 180 kg, that was raised on *Urochloa brizantha* pasture. Buffalo 2 was a 6-year-old male of the Carabao breed (*Bubalus bubalis kerabao*, the swamp buffalo), weighing 500 kg, that was raised in a pen with forage and commercial feed. Both animals had received mineral supplements and had water available ad libitum.

According to the owners, the clinical histories of the buffaloes were very similar, being characterized by progressive weight loss, recurrent tympany, abdominal distention, anorexia, and scant feces, sometimes with a firm consistency and sometimes loose. Both animals were clinically examined [1] and submitted to the atropine test (40 mg atropine sulfate subcutaneously) using the techniques described by Dirksen [1]. Buffalo 1 underwent orogastric intubation, and due to the recurrent tympany, an exploratory laparotomy was performed using the procedures described by Dirksen. [1]. Buffalo 2 underwent a thoracic and abdominal ultrasound examination (Mindray, model Z5 Vet, Shenzen, China) using a 5.0 Mhz convex transducer while in the standing position.

Based on the history, clinical examination, elapsed time of the disease (the period between the onset of clinical signs or clinical suspicion until death in Buffalo 1 was approximately 70 days and in buffalo 50 days) and ultrasound findings, the animals had poor prognoses. Buffalo 2 died (the animal was found dead, and the necroscopic findings showed pneumonia due to aspiration of the rumen contents), and Buffalo 1, by decision of the owner, was euthanized according to the animal welfare guidelines and the rules of the Brazilian National Council for the Control of Animal Experimentation [15]. A necropsy evaluation was performed on both animals.

## 3. Results

The clinical examination of Buffalo 1 revealed apathy, a dull and rough coat, a heart rate of 56 beats per minute (bpm), and a respiratory rate of 26 breaths per minute. Buffalo 2 showed apathy, a lack of appetite, a heart rate of 42 bpm, a respiratory rate of 16 breaths per minute, and scant feces. The two buffaloes had ruminal hypermotility with an average of ruminal movements (RM) of 4–5 RM/minute and ruminal tympany. Palpation revealed no stratification of the rumen contents. Furthermore, both animals (Figure 1) had abdominal distension involving the whole left side of the abdomen and the right ventral quadrant of the abdomen, giving the left side an “apple-shaped” appearance and the right side a “pear-shaped” appearance when viewed from behind. When considered jointly, these shapes form a “papple shape”.

A large amount of gas was removed from Buffalo 1 after the orogastric intubation. There was recurrence of tympany. An exploratory laparotomy was performed in the same animal, in which the presence of homogeneous rumen content with a foamy appearance, no stratification of the rumen content, and a reduction in the sizes of the omasum and abomasum were observed. The animal was followed up for another two months after surgery, and the clinical signs described above persisted.

Buffalo 2 had an increase in volume in the right ventrolateral region (Figure 2a), measuring approximately 5 cm in diameter. There was evidence of dilation of the omasum, abomasum, and gallbladder, and a segment of the pylorus was shown to be adhered to the eventration by ultrasonography (Figure 2b). About 14.6 g of fiber was recovered per 100 g of feces from this animal, a value higher than the 6.7 g recovered from a clinically healthy animal of the same age and breed, fed with the same type of grass, and kept in a stall next to the animal in the study. Fiber recovery was carried out using a sieve with a 2 mm screen. Then, the fiber was dried in an oven at 65 °C for 24 h. In Buffalo 2, it was also observed that the fiber was insufficiently decomposed, indicating poor digestion.

Both animals gave positive results for the atropine test, which resulted in increases in their baseline heart rates by 21.4% (68 bpm) in Buffalo 1 and 25% (53 bpm) in Buffalo 2. The necropsy evaluation of Buffalo 1 showed dilation of the esophagus, rumen, and reticulum (Figure 3a) and size reductions for the omasum and abomasum (Figure 3a). The ruminal contents of animal 1 were olive green and foamy with bubbles within the ingesta (Figure 3b). On the other hand, distention of the forestomach and abomasum was identified in Buffalo 2 (Figure 4a); the complex rumen–reticulum and omasum contents were semi-liquid (solid intermixed with liquid) and had a yellowish color (Figure 4b). In animal 2, there was adherence to the pyloric region in the eventration region (Figure 4c), corresponding to the increase in volume seen in the clinical examination.

## 4. Discussion

Based on the history, clinical signs, ultrasound examination, atropine test, and necropsy findings, it was possible to diagnose proximal vagal indigestion in Buffalo 1 and distal vagal indigestion in Buffalo 2, these being the first reports of these conditions in this species in the Amazon biome. There is a scarcity of data regarding vagal indigestion in buffaloes, with the case in the literature being that described by Hussain et al. [14], who found a cloth obstructing the cardia, resulting in vagal indigestion in a buffalo. In view of this, the present study used case reports in cattle for the discussion [1,3,4,11,13].

The diagnosis of vagal indigestion was performed in two buffaloes, a young female (Buffalo 1) weighing 180 kg and an adult male weighing about 500 kg (Buffalo 2) from the Murrah and Carabao breeds, one raised in an extensive system and the other in an intensive system. Hoflund’s syndrome has been reported in cattle of both sexes and in individuals of different weights and ages, in European cattle breeds, their crossbreed’s offspring, and miniature cattle, as well as in cattle raised in both semi-intensive and intensive systems [5,6,7,8,9,10,11]. Reis et al. [12] reported the occurrence of the disease in extensively reared Nellore cattle, which shows the possibility of the disease occurring in different rearing systems, as well as in cattle of varying ages, sexes, and weights, as observed in the present study.

In both Buffalo 1, which had proximal vagal indigestion, and Buffalo 2, which had distal vagal indigestion, clinical signs of apathy, anorexia, weight loss, dehydration, dull and rough coats, and scant feces with a firm consistency were observed. This is in accordance with previous studies on cattle [1,2,4,5,6,16], demonstrating that there are no differences in these clinical signs between the two main species of large ruminants used for meat and milk production.

Abdominal distention was observed in the upper and lower left quadrants and in the lower right quadrant (the “apple and pear” shape) in the two buffaloes with vagal indigestion in the present study, which is a characteristic finding of this disease [1,3,4,11,13] The “apple and pear” shape is due to a progressive increase in the rumen volume due to the blockage in the flow of the ruminoreticular ingesta into the omasum, in the case of proximal vagal indigestion, and to the blockage of ingesta in the abomasum in the case of distal vagal indigestion, as found in buffaloes 1 and 2.

The average number of ruminal movements (RM) was 4–5 RM/2 min, which is outside the normal value of 2–3 RM/2 min cited by Dirksen [1]. The ruminal hypermotility and bradycardia presented by the two buffaloes can be explained based on Dirksen [1], Fubini and Divers [4] and Vaughan [3] as an interruption of the vagus nerve, which causes stimuli in the centripetal direction to be transmitted to the central nervous system. Through the efferent fibers, they reach the rumen and the heart, leading to hypermotility and vagotonic bradycardia.

According to Curtis and Groot [5], Dirksen [1], and Fubini and Divers [4], the difference in the dilation of the forestomach and abomasal components between cases of proximal and distal vagal indigestion is related to the site of the vagus nerve. When a lesion of the ventral vagal trunk occurs between the reticulum and the omasum, it can lead to a proximal stenosis, inhibiting the passage of food from the reticulum through the omasum, which causes dilation of the rumen and reticulum and reductions in the omasum and abomasum, characteristics similar to those found in the necropsy exam of Buffalo 1 in this study. When the lesion occurs at the exit of the abomasum, it leads to a stenosis with dilation of the forestomach and abomasum due to the inhibition of abomasal flow, characteristics similar to those observed in the necropsy exam of Buffalo 2. According to Radostits [16], the pasty and foamy content in the forestomach, as observed in the buffaloes in this study, is related to the increased digestion time in the rumen–reticulum complex. On the other hand, according to the same author, the greater quantity of insufficiently decomposed vegetable fibers, characterizing poor digestion, is related to alterations in the motility of the rumen and reticulum.

A megaesophagus was observed in the necropsy exam in Buffalo 1, which is probably the cause of proximal vagal indigestion in Buffalo 1, as it would have resulted in damage to the vagus nerve. This is in line with what was observed by Dirksen [1] and Vaughan [3], who cited pathologies of periesophageal and paraesophageal pathological processes and esophageal diverticula as being responsible for damage of this nature. On the other hand, the cause of distal vagal indigestion in Buffalo 2 is related to adherence of the pylorus segment in the eventration with interruption of nervous impulses from the ventral trunk of the vagus nerve in this region. In addition, there are several diseases that can cause damage to the vagus nerve in cattle, including traumatic reticuloperitonitis, reticular adhesions, leukosis, tuberculous diseases, hemangiomas, and lymphosarcomas [1,4,5,11].

The diagnosis of vagal indigestion should be based on the animal’s history, clinical signs, ultrasonographic findings, the atropine test, and necropsy findings. The clinical examination of Buffalo 1 revealed a heart rate of 56 bpm, while Buffalo 2 had a heart rate of 42 bpm, with normal values being 65 to 80 bpm according to Dirksen [1]. In the atropine test, increases in the baseline heart rate by 21.4% (68 bpm) in Buffalo 1 and 25% (53 bpm) in Buffalo 2 were shown. These values are higher than 16%, which is considered the limit for a positive result in the atropine test [1,17]. It is important to emphasize that it is generally difficult to establish the cause of nerve damage with a clinical exam only. This often occurs only at necropsy, demonstrating its importance [2,3,8,10,11]. The necropsy exam of Buffalo 2 confirmed the dilation of the omasum and abomasum, biliary insufficiency, and the presence of the pylorus segment in the herniated sac observed in the ultrasound scan. This indicates that ultrasound is an extremely important and valid exam for the diagnosis of vagal indigestion. However, this disease presents with progressive clinical signs and a poor prognosis, which often makes it necessary to euthanize the animal, as was done in the present study [1,2,4,5,11].

## 5. Conclusions

Given the findings observed in the present study, it can be concluded that vagal indigestion can affect buffaloes of either sex, regardless of age, weight, breed, or production system. Clinical and ultrasonographic examinations associated with the necropsy findings and the atropine test are important tests for confirming this diagnosis in this species. In addition, more reports of the disease are required in order to build a solid body of information about the occurrence and importance of vagal indigestion in *Bubalus bubalis*.

## Figures and Tables

**Figure 1 vetsci-10-00254-f001:**
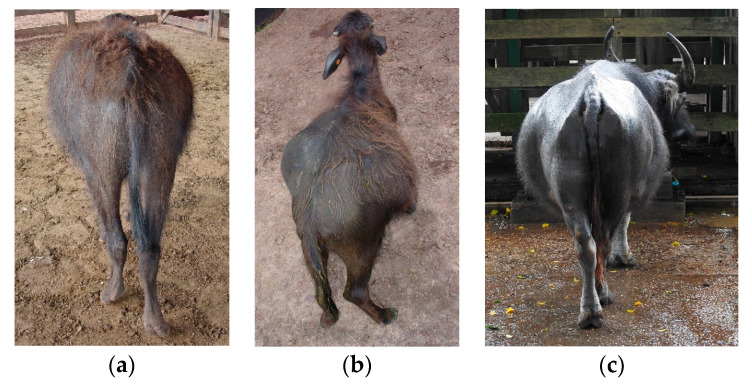
Bilateral distension of the abdomen (“papple shape”): (**a**,**b**) Buffalo 1; (**c**) Buffalo 2.

**Figure 2 vetsci-10-00254-f002:**
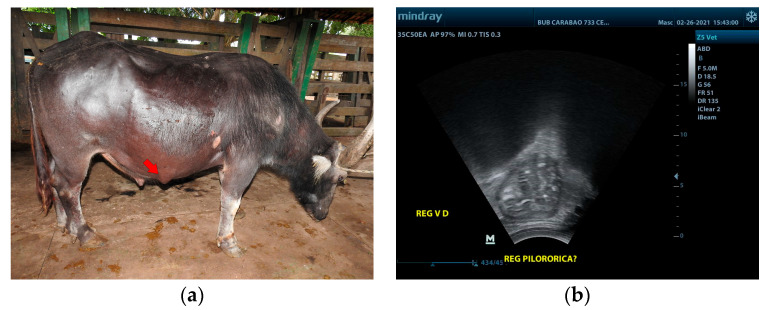
Distal vagal indigestion in Buffalo 2: (**a**) abdominal distention with eventration (arrow); (**b**) segment of the pylorus adhered to the eventration on the ultrasonograph.

**Figure 3 vetsci-10-00254-f003:**
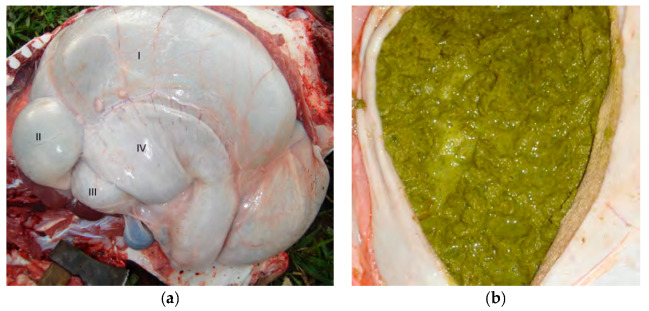
Necropsy findings showing proximal vagal indigestion in Buffalo 1: (**a**) dilation of the rumen and reticulum (I and II); reductions in the size and content of the omasum and abomasum (III and IV); (**b**) foamy and olive-green ruminal contents.

**Figure 4 vetsci-10-00254-f004:**
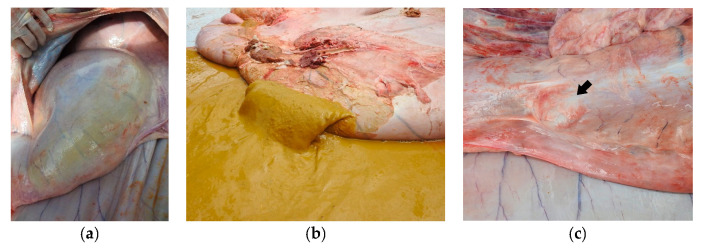
Distal vagal indigestion in Buffalo 2: (**a**) Bilateral distention of the abomasum; (**b**) Semiliquid (solid intermixed with liquid) and yellowish-color ruminal contents; (**c**) Region of the pylorus in which the area was removed from the eventration (arrow).

## Data Availability

Not applicable.

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
