# Peer review of "Proximal and Distal Vagal Indigestion in Buffaloes (Bubalus bubalis) in the Amazon Biome"

_vetsci, 2023, doi:10.3390/vetsci10040254_

Round 1

Reviewer 1 Report

Te work provide a good contribution to the knowledge of Hoflund syndrome/phorestomach  pathology and vagal sndrome in buffalo. The study deserve to be published to fill a gap that existed on the subject.

Author Response

Dear reviewer 1, we thank you for your comments and suggestions, and inform you that the article has been sent for review and extensive editing of the English language has been carried out as per your request. I attach the certificate issued by the service provided by MDPI English Editing.

Reviewer 2 Report

The medicine of bufalloes does note have the quantity and quality of information that exists in bovine medicine. There a few reports on vagal indigestion in buffaloes medicine. In this article, cases are reported in great medical detail. 

1- I suggest inserting reference values on the cardiac rate of bufalloes in the text. Also the results of atropine test.

2- Line 88 - rumen - Incomplete moviments. Information I think is inappropriate. Maybe just righ ruminal frequency. (Cite the reference values of ruminal frequency and source.)

3- Put in the text the base reference of the clinal examination. Maybe Dirksen et al. 

Author Response

Dear reviewer 2, we thank you for your comments and suggestions, and we inform you that we have inserted the reference values on the cardiac rate of buffaloes in the text. Also the results of atropine test. We also remove the expression Line 88 - rumen - Incomplete movements and insert reference values of ruminal frequency and source. And finally we insert base reference of the clinical examination that fori Dirksen (2005). Therefore, all his suggestions were carried out and included in the text. Additionally, we inform you that the article was sent to the English improvement service, as per the attached document from MDPI English Editing.

Reviewer 3 Report

Dear authors,

Buffalo breeding is very important in the Amazon biome and many of its characteristics and adversities should be studied. On the other hand, the Hoflund syndrome is well-known by all ruminant practitioners. Although few reports of the disease in Buffalos are found in the literature, the cases like the ones described are not a novelty at all. 

In this reviewer's opinion, the great finding is the chronic distal stenosis in buffalo 2, which generally occurs in an acute form.

Title and terms

The most used terms for “anterior" and “posterior" vagal indigestion are “proximal" and “distal" vagal indigestion, respectively. 

Material and methods

Was a blood sample collected? If yes, which were the exams performed?

Although the authors describe that the two cases were similar in their clinical history, with progressive weight loss, in the images the animals appear to be in good corporal condition and weight for their age. The authors could be more specific in describing the elapsed time of the disease. What was the cause of death of buffalo 2? And, again, how long did it take from initial symptoms to death?

Line 66: “180kg live weight”, if the animal was alive, there is no need to put “live”. Please change by "weighting 180kg”.

Line 68: there is no need for the term “live weight”. What do “bulk feed” and “supply of concentrate" mean in this context?

Line 69: …mineral “supplement”

Line 82: … “in view of this”?

Results

Judging by the images showed, buffalo 2 seems to have a “paple" appearence or a "pear and apple shape”.

Line 90: “bolf”? Did the authors mean “both”?

Discussion

Bufalo 1 did not have a low heart rate, based on its age and weight, nor a low respiratory rate. On the other hand, buffalo 2 had both low, heart and respiratory rates. The authors said buffalo 1 had megaesophagus at necropsy, but did the animal show any other signs of it before?

Lines 155 and 156: Both animals showed apathy, anorexia…

Line 170: heart nervous system

Conclusions

What different etiology was observed in these cases? Megaesophagus causing vagal damage? Or pyloric incarceration? None of them are different. 

Line 214: “race”?

This reviewer does not think there is any novelty in this manuscript. These reported cases are relatively common in the ruminant practitioner routine. And the English language should be reviewed.

Author Response

Dear reviewer 3, we appreciate your comments and suggestions, and we inform you that the great finding is the chronic distal stenosis in buffalo 2 was inserted in the article, in the Simple Summary, highlighting this information.

Title and terms
Throughout the entire text, the expressions “anterior" and “posterior" were changed to the correct terms “proximal" and “distal", respectively.

material and methods
No blood was collected from the animals. Line 66, Line 69, and Line 82 corrections were performed.
Although buffalo 2 appears to be in good body condition and weight for their age in the photo, we inform you that the photo was taken from an angle that did not favor showing the real body condition of the animal, the photo demonstrates much more the papple shape of the buffalo 2. This animal had a clinical period of the disease of approximately 50 days and ended up being found dead. the animal's breathing, in addition to causing regurgitation of the rumenal content, leading to aspiration pneumonia and death of the animal. If the reviewer is of interest to the reviewer, we can send the photos that demonstrate the above report.

Results
The expression both in line 90 was corrected, in addition to a new description, as per your suggestion, of the papple shape of buffalo 2. the abdomen, giving the left side an 'apple shaped' appearance and the right side a 'pear shaped' appearance when viewed from behind. When considered jointly, these shapes form a ‘papple shape’.

Discussion
Correction was performed on line 170, the correct one was central nervous system. Buffalo 1 had megaesophagus at necropsy, but the animal had not previously shown any other signs. And we affirm that both animals showed apathy, anorexia and low heart and respiratory rates. The period between the onset of clinical signs or clinical suspicion until death in buffalo 1 was approximately 70 days and in buffalo 50 days.

Conclusions
The correction was made on line 214 and, according to its perfect questioning, the question of different etiologies was removed. The conclusion is as follows: "Given the findings observed in the present study, it can be concluded that vagal indigestion can affect buffaloes of either sex, regardless of age, weight, breed, or production system. Clinical and ultrasonographic examinations associated with the necropsy findings and the atropine test are important tests for confirming this diagnosis in this species. In addition, more reports of the disease are required in order to build a solid body of information about the occurrence and importance of vagal indigestion in Bubalus bubalis."

Additionally, we inform you that the article was sent to the English improvement service, as per the attached document from MDPI English Editing.

Round 2

Reviewer 3 Report

Dear Authors, 

Thankyou for your answer.

Still, the information you gave in the answer about the clinical history (p.e. onset of clincal signs to death, regurgitation of rumenal content) and necroscopic findings (p.e. rumenal content in the lungs?) shoud be inserted in the text.

At least I got unhappy that there was no blood colection, that information could take this manuscript to other level.

Author Response

Dear reviewer 3,
once again we appreciate his considerations and inform you that all his requests have been inserted in the text of the manuscript in lines 86-90.

We regret that we did not collect blood samples as instructed. This will be a learning experience and we will avoid making such a mistake in future work.

We also modified the bibliographic references, remaining only those relevant to the manuscript.

I send the attached article so you can check all the changes. And we reiterate our thanks for your willingness to review the article and indicate new improvements.
